# PSHuman: Photorealistic Single-view Human Reconstruction using Cross-Scale Diffusion

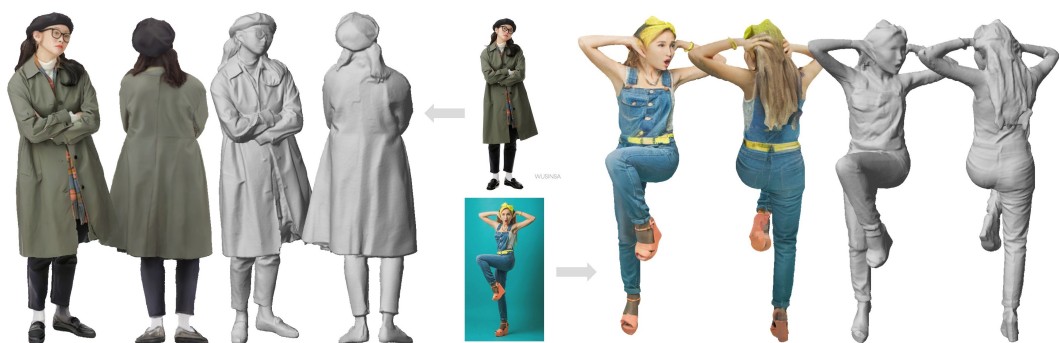

Figure 1: We introduce PSHuman, a diffusion-based full-body human reconstruction model. Given a single image of a clothed person, our method facilitates detailed geometry and realistic 3D human appearance across various poses within **one minute**.

## Abstract

Detailed and photorealistic 3D human modeling is essential for various applications and has seen tremendous progress. However, full-body reconstruction from a monocular RGB image remains challenging due to the ill-posed nature of the problem and sophisticated clothing topology with self-occlusions. In this paper, we propose **PSHuman**, a novel framework that explicitly reconstructs human meshes utilizing priors from the multi-view diffusion model. It is found that directly applying multiview diffusion on single-view human images leads to severe geometric distortions, especially on generated faces. To address it, we propose a cross-scale diffusion that models the joint probability distribution of global full-body shape and local facial characteristics, enabling detailed and identity-preserved novel-view generation without any geometric distortion. Moreover, to enhance cross-view body shape consistency of varied human poses, we condition the generative model on parametric models like SMPL-X, which provide body priors and prevent unnatural views inconsistent with human anatomy. Leveraging the generated multi-view normal and color images, we present SMPLX-initialized explicit human carving to recover realistic textured human meshes efficiently. Extensive experimental results and quantitative evaluations on CAPE and THuman2.1 datasets demonstrate PSHuman's superiority in geometry details, texture fidelity, and generalization capability. Project page: *https://anonymous.4open.science/w/pshuman_anonymous-027F/*.

## 1 Introduction

Photorealistic 3D reconstruction of clothed humans is a promising and widely investigated research domain with significant applications across several industries, including gaming, movies, fashion, and AR/VR (Ma et al., 2021; Orts-Escolano et al., 2016). Traditional methods, which perform multiview stereo and non-rigid registration using multi-camera setups or incorporate additional depth signals, have achieved accurate modeling. However, reconstruction from an in-the-wild RGB image remains an open problem due to sophisticated body poses and complex clothing topology.

A plethora of studies have been developed to address these challenges. PIFu (Saito et al., 2019) and related efforts (Saito et al., 2020; Zhang et al., 2024b; Ho et al., 2024; Zhang et al., 2024a; Xiu et al.,

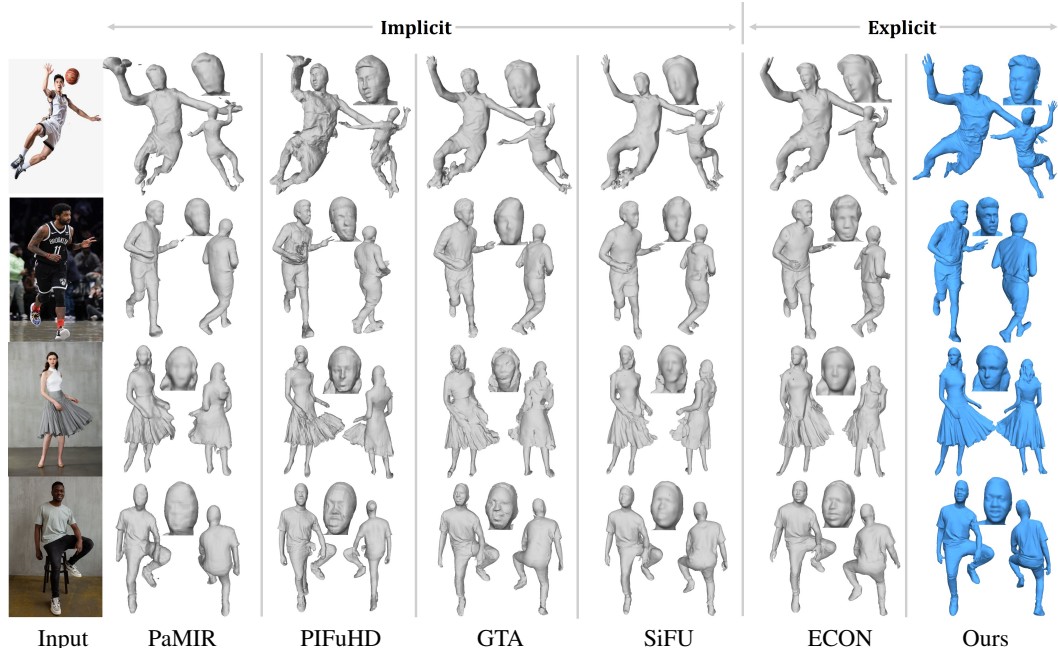

Figure 2: Geometry comparison between **Implicit** and **Explicit** methods.

2022) extract pixel-aligned features from the color or normal image and leverage implicit functions to predict the occupancy field (Mescheder et al., 2019) of the 3D human body and ECON (Xiu et al., 2023) utilizes bilateral normal integration (BiNI) to lift normal clues to 3D body to remain predicted details explicitly. On the one hand, these efforts indeed lead to improvements in terms of either monocular ambiguity or postural intricacy through the introduction of other geometric clues or occluded-view information. On the other hand, the direct regression paradigm still falls short in detail loss and artifacts. Similarly, recent progress in appearance reconstruction (Zhang et al., 2024b; Ho et al., 2024) follows the implicit function to infer full-body texture, struggling with texture unrealism due to poor generalization capability.

In this study, we aim to tackle these existing challenges by introducing a multiview diffusion model and a normal-guided explicit human reconstruction framework. We build upon the recent progress of diffusion-based multiview generation models to explore their hallucination capabilities for robust human modeling. As depicted in Fig. 4, PSHuman takes a full-body human image as input, followed by a carefully designed multiview diffusion model and an SMPLX-initialized mesh carving module, outputting a textured 3D human mesh.

Specifically, we fine-tune a pre-trained text-to-image diffusion model (such as Stable Diffusion (Rombach et al., 2022b)) to generate multiview color and normal maps conditioned on the input reference. Despite impressive generative performance, this base framework faces two major challenges: 1) **Unnatural body structures**, where diffusion models struggle to generate reasonable novel views of posed humans, often resulting in disproportionate body proportions or missing body parts. This issue arises from the severe self-occlusion in the posed human image and lack of body prior for generative models. To address this, we propose an SMPL-X conditioned diffusion model,

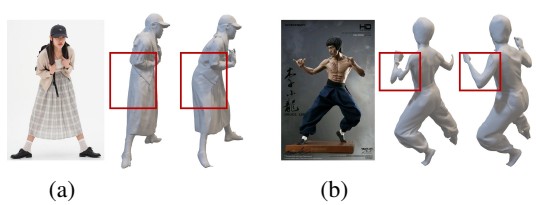

Figure 3: Each triplet contains input (left) and reconstructions of w/o (middle) and w/ (right) SMPL-X condition. Compared with naive diffusion, SMPL-X prior guides handling self-occlusion and improving consistency.

which concatenates renderings of estimated SMPL-X with the input image to provide pose guidance for novel-view generation. This approach constrains the diffusion model to generate consistent

views that adhere to human anatomy, even when fine-tuning with as few as $3,000$ human scans. 2) **Face distortion**, where pre-trained diffusion models often produce distorted and unnatural face details, especially for full-body human input. This problem is attributed to the small size of the face in full-body images, which provides limited information for detailed normal prediction after VAE encoding. To accurately recover face geometry, we propose a body-face cross-scale diffusion framework that simultaneously generates multiview full-body images and local face ones. We also employ a simple yet efficient noise blending layer to enhance face details in global image, guaranteeing both cross-scale and cross-view consistency. Consequently, PSHuman generates high-quality and detailed novel-view human images and corresponding normal maps.

To fully leverage the generated multiview images, we present an SMPLX-initialized explicit human carving module for fast and high-fidelity textured human mesh modeling. Unlike implicit functions that use Multilayer Perceptrons (MLPs) to map normal features to an implicit surface, or BiNI (Cao et al., 2022) that utilizes variational normal integration to recover 2.5D surfaces, we directly reconstruct the 3D mesh supervised by generated multiview normal maps. In practice, we initialize the human model with predicted SMPL-X, and deform and remesh it with differentiable rasterization Palfinger (2022). As shown in Fig. 2, PSHuman can preserve fine-grained details, such as facial features and fabric wrinkles, and generate natural and harmonious novel views. For texturing on the generated meshes, we first fuse multiview color images using differentiable rendering to mitigate generative inconsistencies, then project them onto the reconstructed 3D mesh.

The entire reconstruction process takes as few as one minute. It is noted that recent SDS-based methods (Huang et al., 2024b;a) also achieve state-of-the-art performance in geometry details and appearance fidelity. However, they can only handle simple poses and suffer from time-consuming optimization (such as TeCH Huang et al. (2024b), which takes approximately six hours). Conversely, PSHuman achieves a balance between precision, efficiency, and pose robustness.

In summary, our key contributions include:

- We introduce PSHuman, a novel diffusion-based explicit method for detailed and realistic 3D human modeling from a single image.
- We present a body-face cross-scale diffusion and an SMPL-X conditioned multi-view diffusion for high-quality full-body human image generation with high-fidelity face details.
- We design an SMPLX-initialized explicit human carving module to fast recover textured human mesh based on generated multi-view cross-domain images, achieving SOTA performance on THuman2.1 and CAPE datasets.

## 2 RELATED WORK

Single-image human reconstruction has seen rapid advancements in recent years, primarily driven by three key approaches: implicit function-based reconstruction, explicit shape-based reconstruction, and the emerging 2D diffusion-based methods.

**Implicit Human Reconstruction.** Implicit functions have gained significant traction in human reconstruction (Chibane et al., 2020; Gropp et al., 2020; Yang et al., 2023) due to their flexibility in handling complex topology and diverse clothing styles. Pioneering works such as PIFu Saito et al. (2019) introduce pixel-aligned implicit functions, mapping 2D image features to 3D implicit surface for continuous modeling. Building upon this, subsequent research incorporates parametric models (e.g., SMPL) to enhance anatomical plausibility and robustness in challenging in-the-wild poses (He et al., 2020; Xiu et al., 2022; Zheng et al., 2021; Zhang et al., 2024a) or for animation-ready modeling (Huang et al., 2020; He et al., 2021). Other efforts enhance geometric details and dynamic stability by introducing normal (Saito et al., 2020), depth clues (Yu et al., 2021b; Zheng et al., 2023), or decoupling albedo (Alldieck et al., 2022) from natural inputs. However, these methods struggle with unseen areas due to limited observed information. More recent approaches (Zhang et al., 2024b; Ho et al., 2024) incorporate predicted side-view images to enhance visualization but still face challenges in balancing quality, efficiency, and robustness.

**Explicit Human Reconstruction.** Early research focuses on explicit representation for human reconstruction. Voxel-based methods (Varol et al., 2018; Zheng et al., 2019) utilize 3D UNet to predict volumetric confidence occupied by the human body, which demands high memory and often results

Figure 4: **Overall pipeline**. Given a single full-body human image, PSHuman recovers the texture human mesh by two stages: 1) Body-face enhanced and SMPL-X conditioned multi-view generation. The input image and predicted SMPL-X are fed into a multi-view image diffusion model to generate six views of global full-body images and local face images. 2) SMPLX-initialized explicit human carving. Utilizing generated normal and color maps to deform and remesh the SMPL-X with differentiable rasterization.

in compromised spatial resolution, hindering the capture of fine details crucial for realistic representation. As a more efficient alternative, visual hulls (Natsume et al., 2019) approximate 3D shapes by incorporating silhouettes and 3D joints. Another strategy involves using depth (Gabeur et al., 2019; Smith et al., 2019; Han et al., 2023) or normal (Alldieck et al., 2019; Xiu et al., 2023) information to explicitly infer the 3D human body, balancing detail preservation with computational efficiency. Among these, ECON utilizes normal integration and shape completion, achieving extreme robustness for challenging poses and loose clothing. The major limitations lie in sub-optimal geometry and supporting appearance. To address this, we propose to simultaneously recover geometry and appearance with differentiable rasterization under the supervision of multi-view normal and color maps predicted by the diffusion model.

**Diffusion-based Human Reconstruction.** Most recently, Score Distillation Sampling (SDS) Poole et al. (2022) based human generation methods (Liao et al., 2023; Huang et al., 2024b) have achieved SOTA performance. However, these approaches often require time-consuming optimization. Draw inspiration from the advancement of multi-view diffusion based 3D generation (Liu et al., 2023; Long et al., 2024; Li et al., 2024; Voleti et al., 2024; Tang et al., 2024), our work reduces the inference time by directly generating multiple human views for human reconstruction. We further augment human generation capabilities through the introduction of a novel SMPL-X-conditioned cross-scale attention framework. Most related to our work, Chupa Kim et al. (2023) also reconstructs with multi-view normals. However, it still depends on optimization-based refinement and does not support image condition and texture modeling.

## 3 OUR APPROACH

**Overview.** Given a single color image, we aim to reconstruct the textured 3D human mesh with generated realistic invisible views. PSHuman is built upon recent multi-view generative models (Li et al., 2024; Long et al., 2024), including two primary stages: 1) a body-face cross-scale diffusion model conditioned on SMPL-X, which generates multi-view full-body cross-domain (color and normal) images and local facial ones (Sec. 3.1), 2) an SMPLX-initialized explicit human carving module for modeling 3D textured meshes (Sec. 3.2). Since we generate normal maps and images, we use $x$ and $z$ as the raw data and latents for both data modalities.

### 3.1 BODY-FACE MULTI-VIEW DIFFUSION

#### 3.1.1 BODY-FACE DIFFUSION

**Motivation.** Simply adopting the multiview diffusion (Li et al., 2024; Long et al., 2024) for 3D human reconstruction leads to distorted faces and changes of face identities in the reconstruction results. Because the face only occupies a small region with a low resolution in the image and

cannot be accurately generated by the multiview diffusion model. Since humans are very sensitive to slight changes in faces, such generation inaccuracy of faces leads to obvious distortion and identity changes. This motivates us to separately apply another multiview diffusion model to generate the face at a high resolution with more accuracy.

**Forward and reverse processes.** We define our data distribution $p(x)$ as the joint distribution of the human face $x^F$ and the human body $x^B$ by

$$p(\mathbf{x}) = p(x^B, x^F) = p(x^B|x^F)p(x^F). \tag{1}$$

Then, we follow the DDPM model to define our forward and reverse diffusion process by

$$q(x_t|x_{t-1}) = q(x_t^B|x_{t-1}^B, x_{t-1}^F)q(x_t^F|x_{t-1}^F), \tag{2}$$

$$p(x_{t-1}|x_t) = p(x_{t-1}^B|x_t^B, x_{t-1}^F)p(x_{t-1}^F|x_t^F), \tag{3}$$

where $q$ defines the forward process to add noises to the original data and $p$ defines the reverse process to generate data by denoising. For the forward process, we simply omit the condition on the $x_{t-1}^F$ and add noises to the face and body images separately by the approximated forward process

$$q(x_t|x_{t-1}) \approx q(x_t^B|x_{t-1}^B)q(x_t^F|x_{t-1}^F). \tag{4}$$

Although explicitly defining forward process for $q(x_t^B|x_{t-1}^B, x_{t-1}^F)$ is feasible for the vanilla diffusion model, it is difficult for the latent diffusion model. We explain this difficulty and the feasibility of this approximation in Sec. A.1. For the reverse process $p(x_{t-1}|x_t)$, the face diffusion is just a vanilla diffusion model $p(x_{t-1}^F|p_t^F)$ while the body diffusion model will additionally use the face denoising results as conditions by $p(x_{t-1}^B|p_t^B, p_{t-1}^F)$, as shown in Fig. 5, which is implemented by the following joint denoising scheme.

**Joint denoising.** We utilize a simple but efficient noise blending layer to jointly denoise in body-face diffusion. Specifically, in each self-attention block of UNet, we extract the latent vector of the face branch, resize it with scale $s$, and add it to the face region of the global branch with a weighted sum. Specifically, let us take one of the hidden layers as an example. We denote $h_t^{B_n}$ and $h_t^F$ as hidden vectors of the $n$-th body view and face view at the same attention layer [1] and timestep $t$, the blending operation can be written as

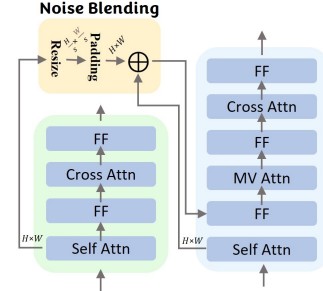

Figure 5: Illustration of joint denoising diffusion block.

$$h_t^{B_n} = \begin{cases} h_t^{B_1} + w \cdot RP(h_t^F, s), & n = 1 \\ h_t^{B_n}, & n = 2, 3, \ldots, N \end{cases} \tag{5}$$

where, the $RP$ is the resize and padding function, $w$ is the mask of the face region. The resulting latent vector can be represented by $z_t^{B_n}$ and $z_t^F$. We jointly optimize the body and face distribution with the following loss,

$$\ell = \mathbb{E}_{t, \mathbf{z}_0^F, \epsilon}\left[\|\epsilon - \epsilon_\theta(z_t^F, t)\|_2\right] + \mathbb{E}_{t, \mathbf{z}_0^B, \mathbf{z}_0^F, n, \epsilon}\left[\|\epsilon^{(n)} - \epsilon_\theta^{(n)}(\mathbf{z}_t^B, \mathbf{z}_t^F, t)\|_2\right], \tag{6}$$

where $\theta$ is shared weights between face and multiple body views. The noise blending allows the face information to be transferred to novel body views with cross-view attention, improving the overall consistency of generated human images.

### 3.1.2 SMPL-X GUIDED MULTI-VIEW DIFFUSION

The diffusion model excels in generating plausible novel views for simple, non-occluded body poses, producing natural human geometry. However, it faces significant challenges with in-the-wild images that often feature self-occlusions. These occlusions can lead to "hallucinations" that violate human structural integrity or exhibit inconsistent limb poses. For example, Fig. 3 illustrates two common issues: (a) the model generating upright side views for a bending posture input, and (b) inconsistencies in arm regions of side views due to self-occlusion, resulting in failed reconstruction.

---

[1]Here, we omit the layer subscript for simplicity.

To mitigate these impediments, we propose incorporating additional pose guidance into the diffusion process. Our method first estimates the SMPL-X parameters of the input image and renders them from six target viewpoints. We then utilize a pre-trained Variational Autoencoder (VAE) encoder to convert these renderings into latent vectors, which are concatenated with noise samples and the reference image to serve as input of the denoising UNet. The introduction of these conditional signals constrains the multi-view distribution, leading to more accurate and consistent human image generation. This approach significantly enhances the model's generalization capability on complex human poses with self-occlusion.

### 3.2 SMPLX-INITIALIZED EXPLICIT HUMAN CARVING

Following the generation of multi-view color and normal images, we elaborate on our proposed SMPLX-initialized human carving module (Fig. 6) to obtain the textured 3D mesh.

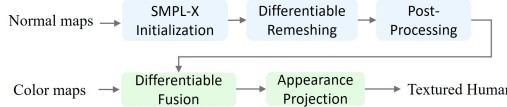

Figure 6: Illustration of our explicit human carving module.

Numerous methodologies have been developed to leverage normal cues for human reconstruction. However, a significant proportion of them employ implicit functions (e.g. MLP) to map the normal feature as implicit surfaces. This process, while effective in certain scenarios, often results in a lack of fine geometric details. Even with BiNI used in ECON, the overall geometry still exhibits a notable degradation. Taking advantage of the multi-view consistent normal maps, we opt to fuse it directly with the explicit triangle mesh. Our reconstruction module consists of three main stages: SMPL-X initialization, differentiable remeshing, and appearance fusion.

**SMPL-X initialization.** The process commences with human mesh initialization, utilizing the aforementioned SMPL-X estimation, which provides a strong body prior, effectively mitigating unnecessary face pruning and densification during subsequent geometry optimization. However, it is noteworthy that the generated multiple views may exhibit slight misalignment with the SMPL model due to normalization and recentering procedures tailored for the diffusion model. Drawing inspiration from ICON, we optimize SMPL-X's translation, shape, and pose by minimizing the pixie-aligned error of multi-view normal and silhouette. The alignment process is computationally efficient, typically requiring only seconds to complete.

**Remeshing with differentiable rasterization.** Given the initial human prior, we utilize differentiable rasterization to carve the details based on observational normal maps. While a common approach involves adding per-vertex displacement to the coarse canonical mesh, this method encounters difficulties when modeling complex details, such as loose clothing. To address this limitation, we directly optimize the SMPL topology, encompassing both vertex positions $V$ and face edges $F$. The optimization procedure iteratively applies vertex displacement and remeshing to the triangle mesh, utilizing the optimizer proposed in (Palfinger, 2022). The optimization objective can be written as

$$\tilde{V}, \tilde{F} = \arg\min_{V,F} \sum_{i=1}^{N} w_i(\|N_i - \hat{N}_i\|_2 + \|S_i - \hat{S}_i\|_2) + \lambda \sum_j (n_j - n_j^{\text{neig}}) \tag{7}$$

where $n_j$ and $n_j^{\text{neig}}$ denote the vertex normal and the average normal of neighboring vertices, The regularization weight $\lambda$ is set to 0.02. We execute 700 optimization steps to achieve optimal performance. Following the mesh optimization, we employ Poisson reconstruction Kazhdan & Hoppe (2013) to complete minor invisible areas, such as the chin. Additionally, we offer the option to substitute the hands with the estimated SMPL-X results (Xiu et al., 2023).

**Appearance fusion.** Upon obtaining the 3D geometry, our objective is to derive the high-fidelity texture matching the reference image. Despite the availability of multi-view images, direct projection onto the mesh results in conspicuous artifacts, arising from the cross-view inconsistency and inaccurate foreground segmentation. To overcome this, we perform texture fusion and optimize the per-vertex color by minimizing the view-dependent MSE loss between the rendered color images and generated ones utilizing differentiable rendering. Finally, we compute a visibility mask and perform topology-aware interpolation to complete the minor unobserved area.

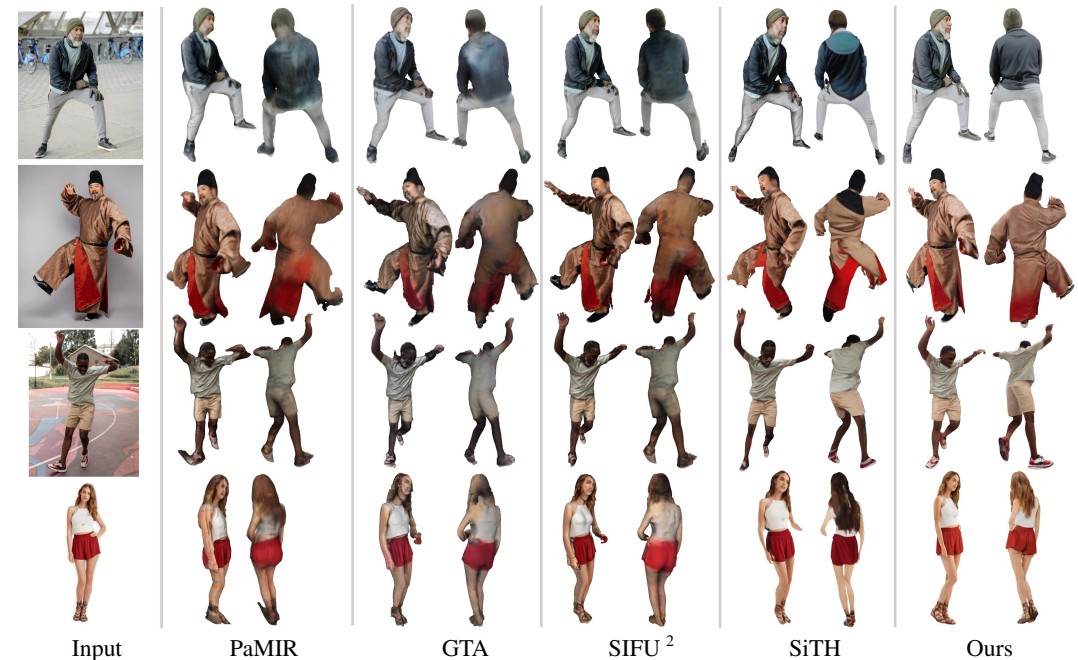

| Input | PaMIR | GTA | SIFU [2] | SiTH | Ours |

Figure 7: Appearance comparisons with methods which produce texture. Our method could reconstruct realistic and reasonable appearance of side and back views.

## 4 EXPERIMENTS

**Dataset.** We conduct experiments on widely used 3D human datasets, including high-quality human scans (THuman2.1 Yu et al. (2021b) and CustomHumans Ho et al. (2023)) captured with a dense DSLR rig and temporal sequence of scans (CAPE Ma et al. (2020)) captured with a body scanner. Specifically, our training dataset comprises $2,385$ scans from THuman2.1 and $647$ scans from CustomHumans. These datasets are selected due to their provision of SMPL-X parameters. For quantitative evaluation, we utilize the remaining $60$ scans from THuman2.1 and $150$ scans from CAPE, with CAPE being subdivided into "CAPE-FP" and "CAPE-NFP" to assess generalization on real-world scenarios. Additionally, we curate a selection of cases from the Internet and SHHQ Fu et al. (2022) fashion data for qualitative comparison.

**Metric.** To assess reconstruction capability, we employ three primary metrics: 1-directional point-to-surface (**P2S**), $L_1$ Chamfer Distance (**CD**), and Normal Consistency (**NC**). CD and P2S quantify the distance between predicted and ground-truth meshes, while NC measures the cosine distance between surface normals. For appearance quality evaluation, we utilize peak signal-to-noise ratio (PSNR), structural similarity index (SSIM), and learned perceptual image patch similarity (LPIPS).

### 4.1 COMPARISONS

**Baselines.** We conducted a comprehensive comparison of our method against state-of-the-art single-view human reconstruction approaches, including PIFu Saito et al. (2019), PIFuHD Saito et al. (2020), PaMIR Zheng et al. (2021), ICON Xiu et al. (2022), ECON Xiu et al. (2023), GTA Zhang et al. (2024a), SiFU Zhang et al. (2024b), and SiTH Ho et al. (2024). For SMPL-based methods, we utilize PIXIE Yu et al. (2021a) for estimation. We also report the results with ground-truth SMPL-X to isolate the impact of pose estimation errors.

**Comparison of geometry quality.** Our method demonstrates superior geometric quality compared to existing approaches, particularly without an SMPL-X body prior (Tab. 1). Unlike template-based methods, which are susceptible to SMPL-X prediction errors, our method supports template-free training, thereby offering enhanced generalization capability. When incorporating the body prior, our method consistently outperforms previous works, demonstrating unprecedented accuracy on

Table 1: Quantitative comparison of geometry quality. To avoid the impact of pose estimation errors on the evaluation, ground-truth SMPL-X models are used during testing. The units for Chamfer and P2S are in cm. The top two results are colored as `first` `second`.

| Method | Publication | CAPE-NFP | | | CAPE-FP | | | THuman2.1 | | |
|---|---|---|---|---|---|---|---|---|---|---|
| | | Cham. Dist ↓ | P2S ↓ | NC ↑ | Cham. Dist ↓ | P2S ↓ | NC ↑ | Cham. Dist ↓ | P2S ↓ | NC ↑ |
| | | | | | w/o SMPL-X body prior | | | | | |
| PIFu | ICCV 2019 | 3.2524 | 2.5469 | 0.7624 | 1.8367 | 1.7582 | 0.8573 | 1.2071 | 1.1299 | 0.7681 |
| PIFuHD | CVPR 2020 | 2.9749 | 2.3677 | 0.7658 | 1.5211 | 1.4834 | 0.8712 | 0.9935 | 0.9647 | 0.7890 |
| PaMIR | TPAMI 2021 | 7.1577 | 3.3832 | 0.6345 | 6.0114 | 3.2877 | 0.6737 | 1.0875 | 1.0144 | 0.7939 |
| ICON | CVPR 2022 | 2.6983 | 2.3911 | 0.7958 | 2.1331 | 2.0359 | 0.8364 | 1.1199 | 1.0925 | 0.7810 |
| ECON | CVPR 2023 | 3.1086 | 2.6044 | 0.7722 | 2.5394 | 2.4336 | 0.8128 | 1.2500 | 1.1469 | 0.7643 |
| GTA | NeurIPS 2023 | 2.7387 | 2.4722 | 0.7875 | 2.2543 | 2.1889 | 0.8247 | 1.0612 | 1.0389 | 0.7857 |
| SIFU | CVPR 2024 | 2.7884 | 2.4792 | 0.7877 | 2.1695 | 2.1107 | 0.8310 | 1.0774 | 1.0586 | 0.7871 |
| SITH | CVPR 2024 | 2.8735 | 2.1226 | 0.7804 | 2.1140 | 1.6754 | 0.8337 | 0.9661 | 0.9034 | 0.7832 |
| **Ours** | - | 2.1625 | 1.6675 | 0.8226 | 1.3615 | 1.1308 | 0.8844 | 0.6609 | 0.5993 | 0.8310 |
| | | | | | w/ SMPL-X body prior | | | | | |
| ICON | CVPR 2022 | 1.5511 | 1.1967 | 0.8572 | 0.9951 | 0.8864 | 0.9190 | 0.6146 | 0.5934 | 0.8493 |
| ECON | CVPR 2023 | 1.8524 | 1.5706 | 0.8392 | 1.1761 | 1.1352 | 0.8969 | 0.6725 | 0.6331 | 0.8362 |
| GTA | NeurIPS 2023 | 1.8853 | 1.4902 | 0.8260 | 1.1484 | 0.9914 | 0.9011 | 0.5791 | 0.5587 | 0.8491 |
| SIFU | CVPR 2024 | 1.5742 | 1.2777 | 0.8529 | 1.0535 | 0.9674 | 0.9024 | 0.5754 | 0.5576 | 0.8500 |
| SITH | CVPR 2024 | 1.8118 | 1.5201 | 0.8345 | 1.1839 | 1.1573 | 0.8870 | 0.6474 | 0.5810 | 0.8264 |
| **Ours** | - | **0.9688** | **0.8675** | **0.8799** | **0.7811** | **0.6984** | **0.9136** | **0.4399** | **0.4077** | **0.8504** |

complex posed humans. The qualitative comparison in Fig. 2 also showcases the superiority of PSHuman, featuring with complete shape, detailed face and natural-looking clothing folds.

Table 2: Quantitative comparison of appearance rendering on THuman2.1 subset.

| Method | PSNR ↑ | SSIM ↑ | LPIPS ↓ |
|---|---|---|---|
| PIFu | 19.3957 | 0.8327 | 0.1001 |
| PaMIR | 19.4130 | 0.8324 | 0.0988 |
| GTA | 19.6071 | 0.8338 | 0.0989 |
| SIFU | 19.4417 | 0.8307 | 0.0985 |
| SITH | 18.4580 | 0.8200 | 0.1004 |
| **Ours** | **20.8548** | **0.8636** | **0.0764** |

Table 3: Evaluation of robustness to SMPL-X estimation on THuman2.1 subset.

| Method | Cham. Dist ↓ | P2S ↓ | NC ↑ |
|---|---|---|---|
| ICON | 0.7827 | 0.6463 | 0.8401 |
| ECON | 0.8022 | 0.6742 | 0.8327 |
| GTA | 0.6631 | 0.6473 | 0.8368 |
| SIFU | 0.6672 | 0.6488 | 0.8302 |
| SITH | 0.6427 | 0.6393 | 0.8241 |
| **Ours** | **0.5574** | **0.5377** | **0.8417** |

**Comparison of appearance quality.** Quantitative evaluations in Tab. 2 reveal that PSHuman outperforms existing methods across multiple metrics, achieving the highest PSNR (20.8548), SSIM (0.8636) as well as the lowest LPIPS (0.0764), which correlates more closely with visual perception. Qualitatively, as illustrated in Fig. 7, PSHuman produces highly consistent appearances on novel viewpoints, including natural and realistic reconstruction for posterior regions. In contrast, existing methods exhibit various limitations such as blurred colors and inconsistent artifacts in unseen views.

**Robustness to SMPL-X estimation.** We assess the robustness of template-based approaches to SMPL-X estimation errors in Tab. 3. Following SIFU, we introduce random noise with a variance of 0.05 to both the pose and shape parameters of the ground-truth SMPL-X model. The results demonstrate the robust reconstruction capabilities of our approach. Furthermore, the efficacy of our method in real-world scenarios is evidenced by the additional results presented in Fig. 13 of A.4.

## 4.2 ABLATION STUDY

**Effectiveness of SMPL-X condition.** In Fig. 3, we show the geometry reconstructed by the models trained without SMPL-X condition and with SMPL-X condition. In Fig. 3(a), it is observed that the naive diffusion model struggles to 'imagen' the pose of a bending human image. Conversely, the SMPL-X provides a strong pose prior to guide the model to generate reasonable side views, leading to better reconstruction. In Fig. 3(b), the diffusion model fails to generate consistent multiple views

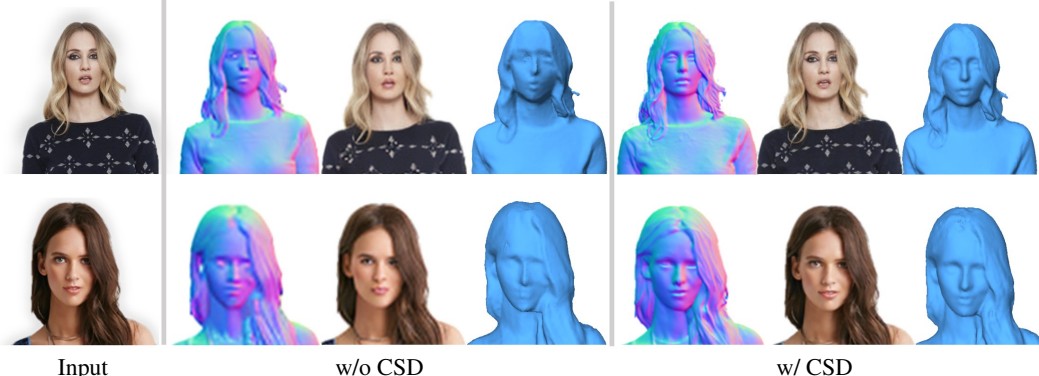

Input            w/o CSD            w/ CSD

Figure 8: Ablation study of the cross-scale diffusion (CSD). The CSD allows sharp face recovery and keeps the identity consistent with the reference input.

due to self-occlusion, resulting in artifacts near the art regions. The SMPL-X guidance effectively enhances consistency, facilitating the complete human body.

**Effectiveness of cross-scale diffusion (CSD).** In Fig. 8, we experiment with the removal of the locally enhanced model, which means only usage of the global diffusion branch. The resulting appearance and geometry, as can be observed, are obviously distorted (e.g. the mouth region) or blurry and fail to accurately recover the consistent geometry details with reference input image. However, using the local enhanced diffusion model, our method manages to overcome these limitations. It achieves more precise and intricate details, contributing to a significant enhancement for the appearance and geometry of 3D humans.

**Effectiveness of mesh carving module.** We assess the efficacy of our reconstruction module by substituting the remeshing step with alternative methods, specifically NeuS and BiNI. As illustrated in Fig. 9, the resulting geometries exhibit notable deficiencies or failures to capture fine geometric details. Note that we employ the normal maps, generated by our diffusion model, across all methods to mitigate potential errors arising from normal prediction discrepancies. Moreover, in the absence of SMPL-X optimization, the reconstructed mesh displays

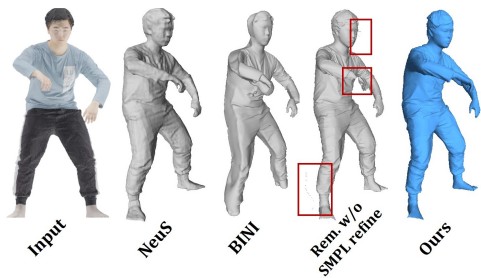

Figure 9: Ablation of our reconstruction module.

subtle artifacts due to misalignment between the initial SMPL-X and the multiple views. Our reconstruction module, which incorporates remeshing with SMPL-X refinement, effectively addresses these issues. For a comprehensive evaluation, we direct the reader to Sec. A.4.

## 5 LIMITATIONS AND CONCLUSION

In this work, we present PSHuman, a single-view human reconstruction framework that significantly enhances the quality of both geometry and appearance. By introducing a body-face cross-scale diffusion model, we improve the capability of modeling high-fidelity 3D human faces. Additionally, we use SMPL-X as guidance for robust multi-view generation. Finally, we devise the multi-view guided explicit human carving module to preserve as many details from generated images as possible. We demonstrate that PSHuman can generate 3D humans with intricate geometric details and realistic appearances, outperforming existing methods.

**Limitations.** We share a common problem with previous template-based works: the pose estimation error has a cascading effect on subsequent view generation and reconstruction. It is promising to mitigate it by unifying existing multi-view datasets and improving the generation robustness without body template conditions.

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

# A APPENDIX

## A.1 DISCUSSIONS ABOUT FACE-BODY CROSS-SCALE DIFFUSION

**Difficulty in implementing dependent forward process**. In the dependent forward process $q(x_t^B | x_{t-1}^B, x_{t-1}^F)$, we know that the face region of $x^B$ corresponds to $x^F$. Since we have defined $p(x_t^F | x_{t-1}^F)$ by adding noises to $x_{t-1}^F$, it is natural to get $x_t^B$ by replacing the pixel values in the face region of $x_t^B$ with $x_t^F$ and just adding noises to the remaining image regions of $x_{t-1}^B$. However, since we adopt a latent diffusion model (Stable Diffusion) Rombach et al. (2022a) here, the pixels of tensors in the latent spaces are not independent of each other so the replacing operation is not valid here. This brings difficulty in separating the face regions in the latent space to explicitly implement the dependent forward process for adding noises.

**Rationale of approximated forward process**. Our rationale for adding noises to the face and the body separately is that the process is similar to multiview diffusion. We can regard the face image and the body image as just two images captured by cameras with different camera positions and focal lengths. In this case, the body-face cross-scale diffusion is a special case of multiview diffusion. In a multiview diffusion, we add noises to multiview images separately so that we can also add noises to the body image and face image separately but consider the dependence in the reverse process.

## A.2 IMPLEMENTATION DETAILS

**Preprocessing.** Our training datasets include scans from THuman2.1 and CustomHumans. For each human model, and the corresponding SMPL-X model, we render 8 color and normal images with alpha channel around the yaw axis, with a $45°$ interval and a resolution of $768 \times 768$. Due to the random face-forward direction, we employ insightface Deng et al. (2018) for face detection, utilizing only viewpoints containing clear facial characteristics for training. As mentioned in the main paper, PSHuman generates 6 color and normal images from front, front-right, right, back, left, and front-left views. To guarantee the generation alignment, we horizontally flip the left and back views during training.

**Diffusion block.** As shown in Fig. 5, our diffusion block consists of two branches, in which the local diffusion inherits from stable diffusion, including self attention, cross attention and feed-forward layers, while the global attention contains an additional multi-view attention layer introduced in Era3D. Global attention is conditioned on the local branch via the alignment of hidden layers.

**Training and evaluation details.** PSHuman builds upon the open-source pre-trained text-to-image generation model, SD2.1-unclip Rombach et al. (2022b). Our training is conducted on a cluster of 16 NVIDIA H800 GPUs, with a batch size of 64 for a total of 30,000 iterations. We adopt an adaptive learning rate schedule, initializing the learning rate at 1e-4 and decreasing it to 5e-5 after 2,000 steps. The entire training process spans approximately 3 days. To enable class-free guidance (CFG) Ho & Salimans (2022) during inference, we randomly omit the clip condition at a rate of 0.05 during training. During inference, we employ PyMAF-X Zhang et al. (2023) for hand pose estimation and PIXIE Feng et al. (2021) for body pose prediction for robustness. For the reconstruction module, we set the number of steps for SMPL-X alignment, geometry optimization, and texture fusion to 700, 100, and 100, respectively, with corresponding learning rates of 0.3, 0.001, and 0.0005. Regarding appearance evaluation, we render color images from four viewpoints at azimuths of $0°, 90°, 180°, 270°$ relative to the input view.

**Inference time.** In Tab. 4, we report the detailed inference time of the whole pipeline, including pre-processing (SMPL-X estimation and SMPL-X image rendering), diffusion, geometry reconstruction (SMPL-X initialization and remeshing) and appearance fusion.

Table 4: Inference time of the reconstruction module.

| Pipeline | Pre-processing | Diffusion | Geo. Recon. | Appearance Fusion |
|----------|----------------|-----------|-------------|-------------------|
| Time / s | 7.2 | 17.6 | 23.3 | 6.0 |

## A.3 USER STUDY

Given the limitations of quantitative metrics in assessing the realism and consistency of side and back views reconstructed from single-view input, we conducted a comprehensive user study to evaluate the geometry and appearance quality of five SOTA methods.

We collect 20 in-the-wild samples and 20 cases from SHHQ fashion dataset for evaluation. Following HumanNorm Huang et al. (2024a), we invite 20 volunteers to evaluate the color and normal video rendered from the reconstructed 3D humans. Participants were instructed to score each model on a 5-point scale (1 being the worst and 5 being the best) across four key dimensions:

- To what extent does the human model exhibit the best geometry quality?
- To what extent does the human model exhibit the best appearance quality?
- To what extent does the novel view's geometry of the human body align with the reference image?
- To what extent does the novel view's appearance of the human body align with the reference image?

Table 5: User study w.r.t reconstruction quality and novel-view consistency.

| Method | PIFuHD | PaMIR | ECON | GTA | SiTH | Ours |
|---|---|---|---|---|---|---|
| Geometry Quality | 1.55 | 1.96 | 3.72 | 2.11 | 2.72 | 4.71 |
| Appearance Quality | - | 1.42 | - | 2.65 | 2.82 | 4.59 |
| Geometry Consistency | 1.69 | 1.76 | 2.48 | 2.33 | 2.79 | 4.61 |
| Appearance Consistency | - | 1.77 | - | 2.16 | 2.73 | 4.68 |

For methods that do not produce texture (PIFuHD and ECON), we only compare the geometry quality and consistency. The results in Tab. 5 indicate that our method represents a significant advancement against SOTA methods, offering superior performance in both geometry and appearance reconstruction, as well as consistency across novel viewpoints.

## A.4 MORE EXPERIMENTS

**Comparison with optimization-based methods.** To assess the efficacy of our approach relative to optimization-based methods, we conducted a comparative analysis of PSHuman against several SDS-based techniques, Magic123, Dreamgaussian, Chupa, and TeCH. Following SiTH, we adopt the pose and text prompt generated by (Li et al., 2022) as condition inputs due to the lack of direct image input support in Chupa. As illustrated in Fig. 10, Magic123 and Dreamgaussian exhibit significant limitations, primarily manifesting as incomplete human body reconstructions and implausible free-view textures. The reliance on text descriptions for conditioning proves insufficient for fine-grained control, resulting in geometries that deviate substantially from the reference inputs. TeCH, a method specifically designed for human reconstruction from a single image, while capable of producing complete human shapes, struggles with severe noise in geometric details and over-saturated textures. These artifacts are characteristic challenges inherent to SDS-based methodologies. In contrast, PSHuman demonstrates superior performance by directly fusing multi-view 2D images in 3D space, enabling the preservation of geometry details at the pixie level while circumventing unrealistic texture. Note that TeCH requires ~6 hours for optimization, PSHuman generates high-quality textured meshes within merely 1 minute.

**Comprehensive quantitative ablation.** In addition to the qualitative ablation in Fig. 3 and Fig. 8, we further conducted comprehensive ablation studies on a subset of 20 samples from the "CAPE-NFP" dataset. Tab. 6 quantitatively illustrates the impact on Chamfer Distance performance when individual components are removed or replaced. It is observed that the SMPL-X condition contributes significantly to reconstruction accuracy. While CSD yields a modest reduction in geometric error, it substantially improves visualization quality and identity fidelity, as evidenced in Fig .8. Furthermore, our reconstruction method, which employs SMPLX-guided differentiable remeshing, demonstrates superior reconstruction performance compared to the BiNI and inpainting pipeline utilized in ECON. The overall results showcase the efficacy of each component in achieving high-quality 3D human reconstruction.

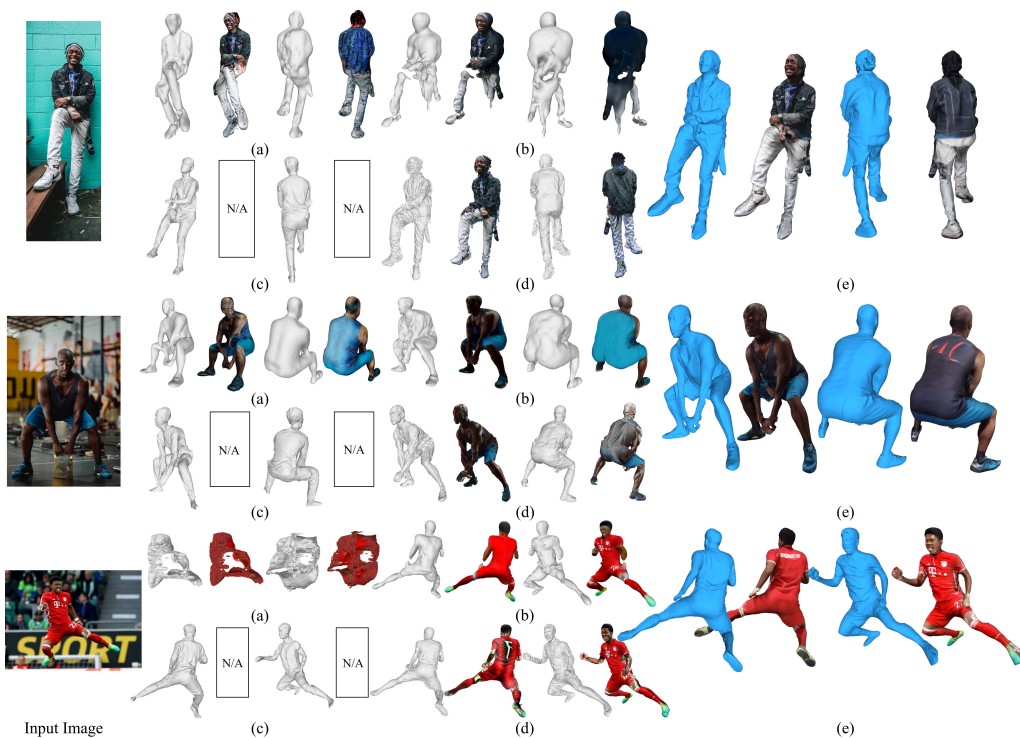

Figure 10: Qualitative comparison with optimization-based methods. We demonstrate the results of **(a)** Magic123, **(b)** Dreamgaussian, **(c)** Chupa, **(d)** TeCH and **(e)** Ours.

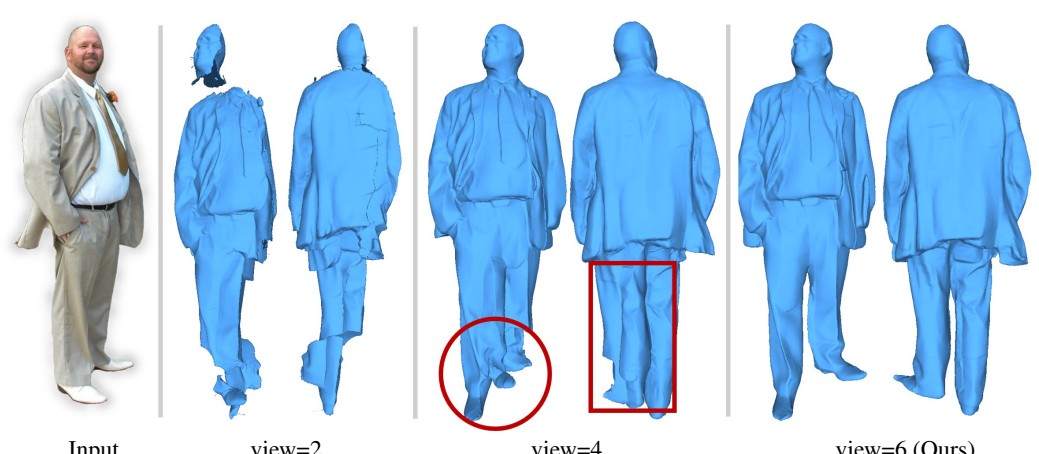

Figure 11: Ablation of view number. Since normal maps lack depth information, optimizing geometry by only two or four views leads to an incomplete or unnatural human structure.

**Ablation of view number.** In Fig. 11, we present the results reconstructed using only two-view (front and back) or four-view (front, right, back, left) normal maps. Since there is a lack of depth in information, optimizing geometry with fewer views leads to severe artifacts, such as incomplete or unnatural human structures. In contrast, it is evident that the artifacts are reduced when using size views, which demonstrates the effectiveness of our multi-view setting.

Table 6: The ablation study of core designs.

| Diffusion | | Reconstruction | | | CD↓ |
|---|---|---|---|---|---|
| CSD | SMPLX-Cond. | Remeshing | SMPLX-ECON | SMPLX-Remeshing | |
| ✗ | ✗ | ✔ | ✗ | ✗ | 1.4920 |
| ✔ | ✗ | ✔ | ✗ | ✗ | 1.4370 |
| ✔ | ✔ | ✔ | ✗ | ✗ | 1.0938 |
| ✔ | ✔ | ✗ | ✔ | ✗ | 1.2630 |
| ✔ | ✔ | ✗ | ✗ | ✔ | **0.9597** (Ours) |

## A.5 ETHICS STATEMENT

While PSHuman aims to provide users with an advanced tool for single-image full-body 3D human model reconstruction, we acknowledge the potential for misuse, particularly in creating deceptive content. This ethical concern extends beyond our specific method to the broader field of generative modeling. As researchers and developers in 3D reconstruction and generative AI, we have a responsibility to continually address these ethical implications. We encourage ongoing dialogue and the development of safeguards to mitigate potential harm while advancing the technology responsibly. Users of PSHuman and similar tools should be aware of these ethical considerations and use the technology in accordance with applicable laws and ethical guidelines.

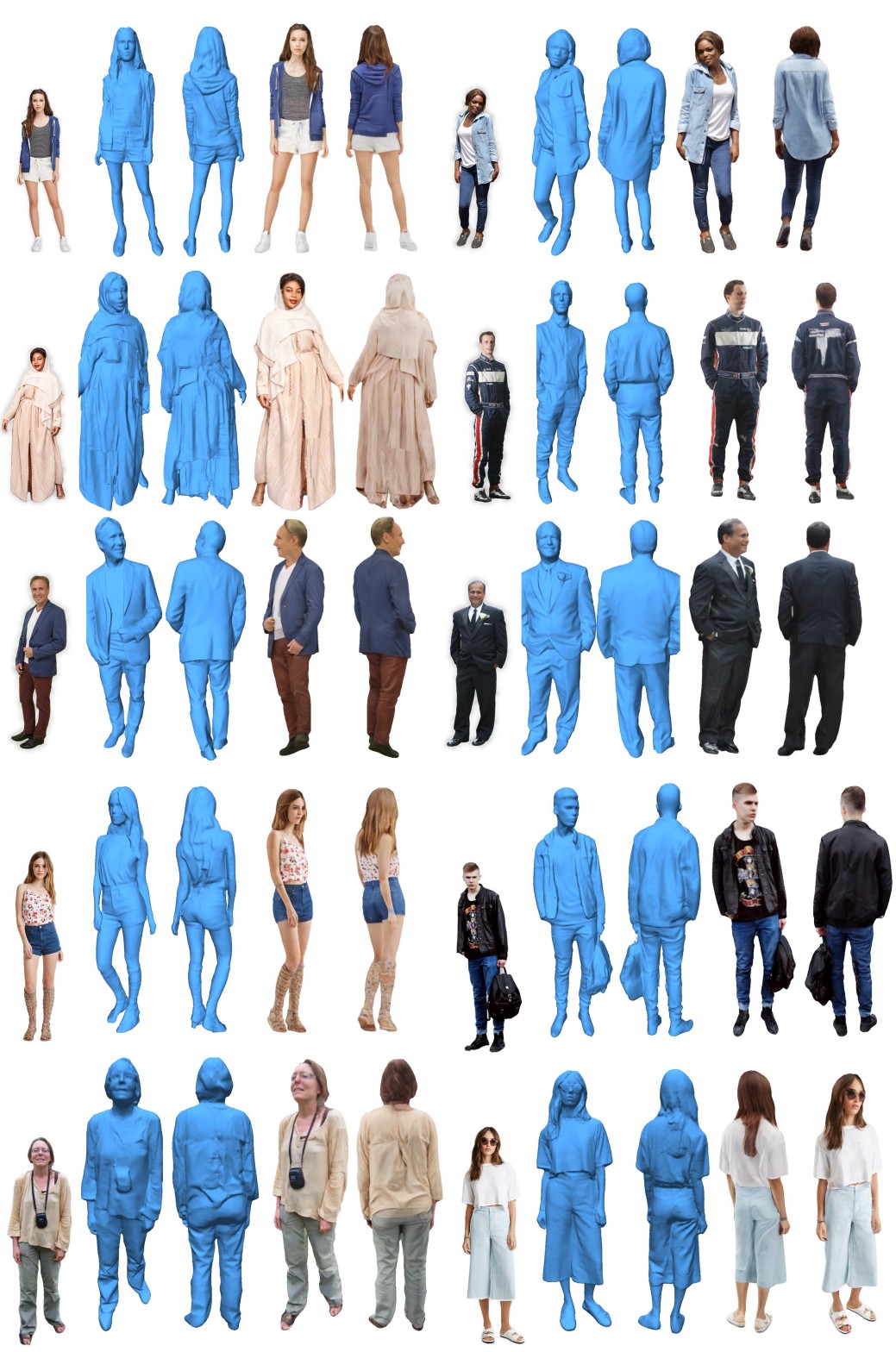

Figure 12: More results on SHHQ dataset.

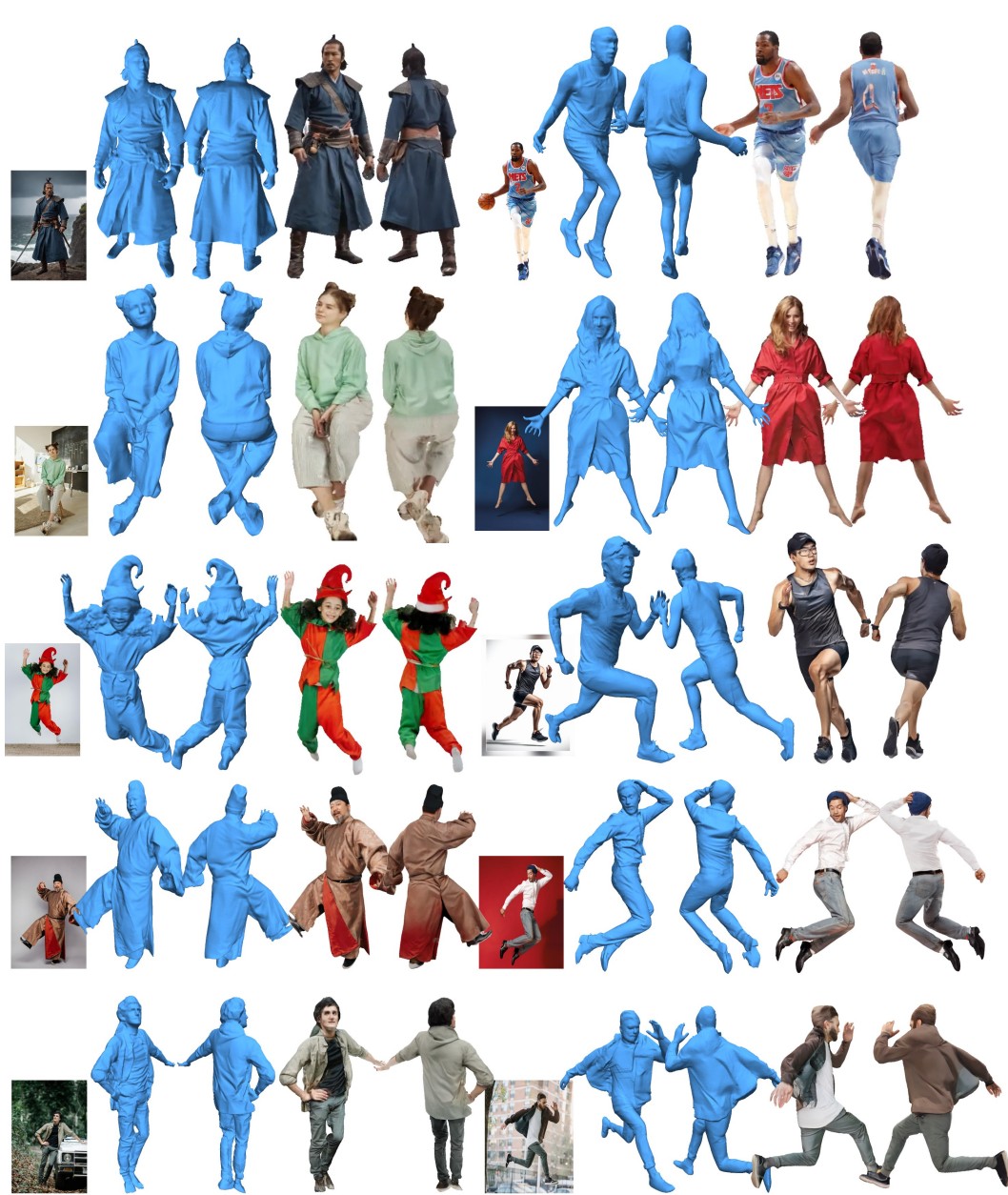

Figure 13: More results on in-the-wild data.

