# OpenReview forum: "PSHuman: Photorealistic Single-view Human Reconstruction using Cross-Scale Diffusion"
_ICLR.cc/2025/Conference — ICLR 2025 Conference Withdrawn Submission_

### Official Review · Reviewer_t5gK · 2024-11-01

**Soundness:** 3
**Presentation:** 2
**Contribution:** 2
**Rating:** 5
**Confidence:** 4

**Summary:**

This paper introduces a single-view human reconstruction approach structured in two stages. In the first stage, multi-view color and normal maps are generated using diffusion models. The process includes a dedicated diffusion model specifically for the face region, whose intermediate features are integrated into a second diffusion model for the full body. In the second stage, the generated color and normal maps from multiple views are aggregated through optimization. The SMPL template is iteratively optimized to fit observations from all views, leading to the final reconstructed output.

**Strengths:**

1. The use of a separate diffusion model for the face allows for detailed and clear facial reconstruction, improving the quality of the facial features in the final model.
2. The approach shows performance improvements over existing single-view human reconstruction methods.

**Weaknesses:**

1.  The approach follows previous works that utilize diffusion models to generate multi-view images as an initial step [1][2][3]. Specifically, in [3], a SMPL-rendering guided control net is also proposed. The main innovation here—a separate diffusion model for the face region—offers improvement but is incremental.

2. An alternative, potentially simpler, method for integrating generated face and body views could be: during the "explicit human carving", optimize shape and color regarding to all 12 views, with some masking approach to optimize face region only based on the generated face views. Will this work?

3. Is a single diffusion network sufficient for both face and full-body multi-view generation? Or they have to be two different models?

4. The explanation of diffusion network is unclear. How is the network conditioned on the input image? Concatenating with noise or by cross attention? How do you initialize the diffusion network? Trained from scratch? How consistent are the generate views? I assume without pose information, the generated poses could be slightly off.

[1] Zero-1-to-3: Zero-shot One Image to 3D Object

[2] Instant3D: Fast Text-to-3D with Sparse-view Generation and Large Reconstruction Model

[3] SiTH: Single-view Textured Human Reconstruction with Image-Conditioned Diffusion

**Questions:**

The design of the joint denoising diffusion network requires more justification to clarify its advantages. Specifically, additional details are needed to explain why this joint approach was chosen over simpler alternatives, such as direct optimization techniques. It would be helpful if the authors could address the concerns raised in Weaknesses 2, 3, and 4.

Additionally, I recommend that the authors revise the paper to include more detailed explanations of these design choices, such as how the diffusion networks are employed.

---

### Official Review · Reviewer_D5kG · 2024-11-01

**Soundness:** 2
**Presentation:** 2
**Contribution:** 2
**Rating:** 3
**Confidence:** 4

**Summary:**

The paper proposes a single-view 3D human reconstruction method with higher fidelity and faster inference speed than prior works by leveraging multi-view diffusion models. The method utilizes a prior SMPL-X mesh and a latent diffusion model to generate multi-view consistent human renderings that are then used to refine the SMPL-X mesh. They showcase results on THuman and CustomHumans datasets and show competitive results.

**Strengths:**

Overall, the paper highlights the benefits of multi-view diffusion models for view-consistent realistic renderings of digital humans. By combining it with a strong prior like SMPL-X, they are able to refine the mesh representation to recover high-quality 3D reconstructions just from a single view. Additionally, they showcase how a single diffusion model can be used to generate both highly detailed body information and high-fidelity face details using a noise blending module, which is a common failure mode for prior methods. Finally, this method performs the entire reconstruction within a minute which is much faster than prior approaches.

**Weaknesses:**

- Central claim is not supported: One of the central claims of the paper is that prior work cannot recover highly-fidelity face details which this method can. But the paper fails to support this clam through quantitative evaluations that purely focus on the enhancements in face quality. Metrics in Tab. 1,2,3 entangle both full-body and face reconstruction quality.
- Quality of results: The overall results for this method do not look much better compared to ECON. Disregarding pose estimation, the high-frequency details either seem very exaggerated or similar to ECON's method for the body. Additionally, it is not clear if the method can handle self-occluding views since none of the figures show results for occluded regions. Finally, since SMPL-X is a strong prior that this method leverages, it would be good to consider adding pure SMPL-X mesh as a baseline for comparison.
- Vague writing: Overall, there needs to be more attention to the draft. There are a lot of inconsistencies, errors and missing details which confuses the reader and falls short on communicating the core message.

**Questions:**

- In Fig.1 Row 1, 2, 4, are the hands replaced with the SMPL-X hands? It looks like it was grafted on top of the reconstruction making it look unnatural. Further, L317 mentions the method has an "option" of doing it. It would be better to clarify this in the image caption, if performed.
- In Fig. 4 caption,
  * it mentions that the diffusion model generates "six views of global full-body images and local face images". But the figure just shows one face image as output. This is not clarified in the method.
  * For the face diffusion, how is the SMPL-X face image cropped? This is not mentioned anywhere in the paper.
  * For the input to the face diffusion model, what is concatenated with the SMPL-X face image? There is just one arrow pointing to the concat operation.
- What is the weight w (L244) used for training and inference? Is it a learned value or is it just a binary mask as mentioned in L251? There is no reference to this in the text.
- In Eq. 5, what is the orientation of the first body view? What is the reason behind blending just the "first" body view and none of the other views? Additionally, how are the 6 views chosen? Is it with respect some canonical orientation of the person and how would the method ensure that orientation?
- For table 1, what are the model inputs when not using the SMPL-X prior? And how is the mesh carving performed without it?
- The paper is riddled with typos, grammatical errors and unnatural sounding sentences:
  * L80 remain -> retain
  * L192 Draw -> Drawing
  * L299 pixie -> pixel
  * L430 imagen -> imagine
  * L448 art -> arm
  * L798 pixie -> pixel
- Placement of Fig. 3 is a bit premature since its not referenced until Page 5.
- L742 why is hand pose estimation done? Also in L317, what is the reason for substituting it with the SMPL-X result? Does the method perform worse on hands?
- Is the evaluation protocol mentioned in L745 borrowed from a previous paper? The sparse number of views might be insufficient to characterize the quality of the reconstructed texture color.
- In Fig. 12, why are the color reconstructions horizontally flipped wrt input image?
- Please specify the model implementation details in the main paper.

**Details Of Ethics Concerns:**

No ethical concerns.

---

### Official Review · Reviewer_QuWN · 2024-11-04

**Soundness:** 3
**Presentation:** 3
**Contribution:** 3
**Rating:** 6
**Confidence:** 4

**Summary:**

The paper introduces PSHuman, a diffusion-based method for single-view 3D human reconstruction that generates detailed and photorealistic models. It uses a cross-scale diffusion model for high-fidelity face details and incorporates SMPL-X for pose guidance. PSHuman efficiently produces textured meshes with improved geometric accuracy.

**Strengths:**

Cross-scale diffusion for enhanced facial details.

SMPL-X conditioning for better pose representation.

Efficient generation of detailed 3D human models.

**Weaknesses:**

1.	What if the generated multi-views are not accurate? Will the reconstructed body be affected?
2.	How much time does it take to reconstruct a human, and how does the speed compare with existing methods like ICON[A], HiLo[B], and D-IF[C]?
3.	More baseline methods, such as HiLo[B] and D-IF[C], should be considered to fully demonstrate the effectiveness of the proposed method.
4.	I recommend that the author provide the six views generated by the proposed diffusion model with respect to Figure 7. It would be more straightforward to understand the effectiveness of the proposed method.

**Reference**

[A] Xiu, Yuliang, et al. "Icon: Implicit clothed humans obtained from normals." 2022 IEEE/CVF Conference on Computer Vision and Pattern Recognition (CVPR). IEEE, 2022.

[B] Yang, Yifan, et al. "HiLo: Detailed and Robust 3D Clothed Human Reconstruction with High-and Low-Frequency Information of Parametric Models." Proceedings of the IEEE/CVF Conference on Computer Vision and Pattern Recognition. 2024.

[C] Yang, Xueting, et al. "D-if: Uncertainty-aware human digitization via implicit distribution field." Proceedings of the IEEE/CVF International Conference on Computer Vision. 2023.

**Questions:**

Please refer to the weakness part.

---

### Official Review · Reviewer_m1YH · 2024-11-05

**Soundness:** 2
**Presentation:** 3
**Contribution:** 3
**Rating:** 6
**Confidence:** 4

**Summary:**

Reconstructing 3D human model from a single image is a long-standing problem. This is a very challenging task. For existing methods, they are still very difficult to recover detailed geometry especially for facial regions and garment wrinkles.  To my best knowledge, this work is the first one that enables high-fidelity reconstruction of human face.  To do so, they proposed several novel designs.

**Strengths:**

- Overall, the work is well-motivated and the paper is well-presented.
- As reported in the figures, the results are really visually good.
- All the method designs seem reasonable. Including joint body-face diffusion module for high-fidelity face recon, SMPL-X guided multi-view diffusion, and the SMPL-X conditioned human mesh carving.

**Weaknesses:**

I still have some concerns on the experiments:
1) Lacking an ablative analysis. The current paper discussed the effectiveness of the proposed CSD(cross-scale diffussion). However, the compared method is only removing the locally enhanced model and just using the global fusion branch. To me, there is another baseline, which is to keep the local branch and only discard the noise blending part, with a follow-up fusion part to fuse the output of the local part with the output of the global branch.

2) Lacking quantitative analysis for many ablation studies. It is good to see many important things are discussed in the ablation study part, which includes SMPL-X condition, CSD and mesh carving module. However, only some visual results on a small set are shown.

3) For the experiment on robustness to SMPL-X estimation. Currently, only random noise with a variance 0.05 is added. Although this is just a following of SIFU, more settings are encouraged to include. Because this work heavily relies on the estimated SMPL-X, and the SOTA estimation method is not that good, as known. More experiments are needed to test the robustness. And, adding extra noise on the face part is also needed to check the impact to the local prediction part.

4) Lacking detailed discussions on the failure cases. As mentioned, the results quality relies on the accuracy of the SMPL-X estimation. I am curious if the SMPL-X is of low accuracy, what does the resulted mesh look like? For appearance, it is also commonly known the texture fusion is very challenging. Thus, are there cases where the seems among different view will happen? More discussions are needed.

Although there are many issues, I still appreciate the SOTA results and think the paper is valuable to this area.

**Questions:**

Q1: As known, the optimization of the mesh, starting from SMPL-X to the target one, especially for loose garments, is very challenging. I am very curious how is the robustness of the proposed method. The authors are suggested to visualize the intermediate optimization stage for justification. Like the second example in Fig 7.

Q2: It is mentioned that an optimization-based texture fusion is conducted to solve the cross-view inconsistency, however, it lacks details. How is the formulation of the optimization? What are variables and what are energy terms?

---

### Note · Authors · 2024-11-14

I have read and agree with the venue's withdrawal policy on behalf of myself and my co-authors.